# CaSO_4_ Increases Yield and Alters the Nutritional Contents in Broccoli (*Brassica oleracea* L. Var. *italica*) Microgreens under NaCl Stress

**DOI:** 10.3390/foods11213485

**Published:** 2022-11-02

**Authors:** Wenjing Zeng, Jing Yang, Guochao Yan, Zhujun Zhu

**Affiliations:** 1College of Environmental and Resource Science, Zhejiang A&F University, Hangzhou 311300, China; 2College of Horticulture Science, Zhejiang A&F University, Hangzhou 311300, China

**Keywords:** sulforaphane, salinity, leaf area, glucosinolates, nitrate, broccoli

## Abstract

Broccoli (*Brassica oleracea* L. Var. *italica*) microgreens are rich in various nutrients, especially sulforaphane. NaCl application is an effective method to reduce nitrate content, and to improve sulforaphane content; however, NaCl application is associated with a risk in productivity reduction. Ca application is a well-known approach to cope with salt stress. Thus, we hypothesized that adding CaSO_4_ may mitigate the adverse effects of NaCl stress, and enhance the quality of broccoli microgreens. In this study, we conducted an experiment to investigate the effects of a combined treatment of NaCl and CaSO_4_ on the fresh yield, glucosinolates (GS), sulforaphane, nitrate, and mineral element contents of broccoli microgreens. The results showed that the incorporation of CaSO_4_ into NaCl solution unexpectedly increased the yield of the leaf area. Moreover, the addition of CaSO_4_ ameliorated the decline in GS under NaCl stress, and induced the accumulation of Ca and S. The nitrate content decreased more than three times, and sulforaphane content also decreased in the combined treatment of NaCl and CaSO_4_. This study proposes that the incorporation of CaSO_4_ into NaCl solution increases the yield, and alleviates the unfavorable effects induced by NaCl stress on the quality of broccoli microgreens. This study provides a novel approach for microgreens production.

## 1. Introduction

Cruciferous microgreens are kinds of vegetables that have been reported to be more nutrient-dense compared to their mature counterparts [1]. They are rich in healthy compounds, such as GS, vitamins, and mineral elements [2,3]. As a result of their abundant bioactive compounds, the chemoprotective effects of cruciferous microgreens against human diseases have been reported in numerous studies [4]. These characteristics make cruciferous microgreens satisfy consumers’ interests in healthy diets, and have broad market prospects [4]. Broccoli microgreens have received extensive research and more attention for being rich in glucoraphanin, compared to other cruciferous microgreens in general [5]. The intact glucoraphanin has little biological activity, while its hydrolysate sulforaphane exhibits impressive effects of cancer chemoprevention [6], neuroprotective protection [7], and anti-diabetes effects [8].

Previous studies have shown that NaCl treatment greatly increases the content of sulforaphane [9,10,11] by increasing transcription abundance of the GS hydrolytic enzyme and its cofactor protein. This directs the GS hydrolytic enzyme to produce sulforaphane rather than sulforaphane nitrile [10]. However, vegetables provide not only beneficial compounds, but also antinutrients for humans, such as nitrates. A high intake of nitrates is associated with a risk of carcinogenic nitrosamine formation [12]. NaCl application has been considered to be an effective way to cope with high nitrate accumulation in vegetables [13] such as lettuce [14] and artichoke [15]. This is due to the competition between nitrate and chloride for the same anion channel [16]. Unfortunately, the risk of NaCl application to microgreens and spouts is apparent, such as a reduction in germination and yield [10,17]. Meanwhile, low yields and short shelf life are two important factors that restrict the development of the microgreens industry [18]. The addition of Ca to alleviate NaCl stress has been reported for many crops [19,20,21]. Moreover, the application of Ca is a mineral fortification approach that is used in producing food crops [22]. Otherwise, sulfur is an essential component of GS. In order to elevate the content of GS, sulfur-containing chemicals are commonly applied to cruciferous sprouts [23,24].

Although the alleviating effect of Ca on NaCl stress has been well documented in common fruits and vegetables, little is currently known about its effects on microgreens. Here, we hypothesized that incorporation of CaSO_4_ into NaCl solution could not only enhance the yield, but also improve the quality of broccoli microgreens; hence, we sought to investigate the effects of adding CaSO_4_ in NaCl solution on the yield and the content of GS, sulforaphane, nitrate, and minerals (P, S, Na, K, Ca, Mg, Zn, Fe).

## 2. Materials and Methods

### 2.1. Experimental Design and Treatments

Broccoli (*Brassica oleracea* L. Var. *italica*) seeds (Wenzhou Zhaofeng Seed Co., Ltd., Wenzhou, China) were rinsed with distilled water after being immersed in 1% (v/w) sodium hypochlorite for 30 min, and were then soaked in distilled water for 3 h at 30 °C. Submerged seeds (2 g seeds per tray, 17 cm × 17 cm) were placed in culture trays that were filled with quartz sand. The microgreens trays were placed in a growth chamber with environmental conditions as follows: a photoperiod of 14 h light/10 h dark, at 22 ± 2 °C. Seeds were germinated for 2 days in distilled water for root elongation of the microgreens. Then, two-day-old microgreens treated with distilled water (CK); 80 mmol/L NaCl (Na); 1.39 mmol/L CaSO_4_ (Ca), which is the concentration of a nearly saturated solution that is not precipitated, dissolved at 10–30 °C; and 80 mmol/L NaCl + 1.39 mmol/L CaSO_4_ (NaCa). The solution in each tray was replaced daily. Trays were arranged randomly, and were systematically rotated every day to enhance the uniformity of the light environment. On the ninth day after sowing, microgreens were harvested, immersed immediately in liquid nitrogen, and then stored at −80 °C for sulforaphane and nitrate, or were lyophilized for GS and mineral elements analysis.

### 2.2. Determination of GS

The GS determination procedures were performed as previously described by Zhu et al. [25], with slight modifications. Briefly, freeze-dried samples (50 mg) were boiled with 5 mL of methanol (70%, *v*/*v*), in order to inactivate myrosinase. Then, 2 mL of the supernatant were loaded into a 1 mL DEAE-Sephadex A25 column, and desulfated overnight with 250 μL of aryl sulfatase. The resultant desulphoglucosinolates were eluted with 2 mL of water, and filtered through a 0.22 μm membrane. Separation and detection were performed on a Waters ACQUITY Arc system (Waters Co., Milford, MA, USA) that was equipped with a 2489 UV/Vis detector, using a prontosil ODS2 column (250 × 4 µm, 5 µm, Bischoff, Leonberg, Germany) at 229 nm. Determination was conducted at a flow rate of 1 mL/min in a linear gradient, beginning with 0% acetonitrile for 1 min, reaching 20% acetonitrile at 32 min, and constant 20% acetonitrile for 6 min. Glucotropaeolin (PanReac AppliChem, Darmstadt, Germany) was added to each sample as an internal standard.

### 2.3. Determination of Sulforaphane

The extraction and analysis of sulforaphane used the method as described by Guo et al. [26], with some modifications. Broccoli microgreens (0.5 g fresh shoots) were ground with 1 mL of distilled water. After 3 h of incubation at 37 °C, the microgreens homogenate was extracted three times with 10 mL of dichloromethane. The dichloromethane fraction was dried at 35 °C under vacuum on a rotary evaporator. The residue was dissolved in 2 mL of acetonitrile and through a 0.22 µm membrane filter. The extracts were analyzed using a Waters ACQUITY Arc system (Waters Co., Milford, MA, USA) that was equipped with a 2489 UV/Vis detector, using a prontosil ODS 2 column (250 × 4 µm, 5 µm, Bischoff, Leonberg, Germany). The flow rate was 0.6 mL/min in a linear gradient of 10–60% acetonitrile from 0 to 25 min, reaching 100% acetonitrile at 30 min. The absorbance value was 254 nm. Sulforaphane (Sigma Chemical Corporation, St. Louis, MO, USA) was used as an external standard.

### 2.4. Determination of Nitrate

The nitrate content was determined spectrophotometrically, as described by Cataldo et al. [27]. One gram of fresh sample was ground in 3 mL of distilled water. The extract was centrifuged at 3000× *g* for 10 min, and the supernatant was filtered. Then, 0.1 mL of the supernatant was mixed with 0.4 mL of 5% (*w*/*v*) salicylic acid (in H_2_SO_4_) and 9.5 mL of 8% NaOH. After 30 min, the nitrate content was measured at a wavelength of 410 nm.

### 2.5. Determination of Mineral Elements

Dry sample was digested with mixed acid (V_HNO3_:V_HF_, 1:1) in a microwave oven digestion system (MARS6, CEM Corporation, Matthews, NC, USA) [28]. The concentrations of elements were determined via inductively coupled plasma-atomic emission spectrometry (iCAP 6300 series, Thermo Fisher Scientific Inc., Waltham, MA, USA). The detailed analytical conditions were previously described [29]. The calibration curve (R^2^ > 0.999) was built using a multi-element calibration standard. The elemental recoveries were in the range of 90 to 110%.

### 2.6. Statistical Analysis

Experimental data were expressed in terms of the mean ± standard deviation (SD) of three biological replicates. Origin 9.0 software was used to calculate and draw data. One-way analysis of variance (ANOVA) with Tukey’s test were conducted on data, and *p* < 0.05 was considered significant [30].

## 3. Results

### 3.1. Yield

The NaCl treatment reduced the yield. However, the combined treatment of NaCl and CaSO_4_ had the largest yield and leaf area among treatments, of which fresh yield was about two-fold compared to the control (Figure 1C). The leaf area from the combined treatment was larger than that of other treatments (Figure 1A,B).

### 3.2. Nutritional Characteristic

NaCl treatment led to the lowest value of both the total GS and individual GS contents, especially in glucoraphanin, glucoiberin, gluconapin, and glucoerucin; moreover, the addition of CaSO_4_ was effective in preventing a decline in GS, resulting in no significant changes in glucoiberin, signirin, gluconapin, and neoglucobrassicin contents compared with the control. Moreover, among all treatments, the content of 4-hydroxyglucobrassicin under the combined treatment was the highest. CaSO_4_ treatment increased the total content of GS in comparison with that of the control (Table 1). The results of sulforaphane showed that NaCl treatment induced the highest sulforaphane content, while no significant changes occurred in the other treatments (Figure 2).

The nitrate level in the NaCl treatment and combined treatment was considerably lower than that of other treatments, more than two times lower (Figure 3). Compared to the control, NaCl treatment increased K but reduced Ca, Mg, and Zn concentrations, while the combined treatment increased the content of Ca and S (Table 2).

## 4. Discussion

Data from the literature confirm that NaCl has biofortification effects as eustress [13]. Our research showed that the biofortification effect of NaCl occurred via sulforaphane increase and nitrate decrease in broccoli microgreens. These findings are consistent with the research of Guo et al. [31], who reported that high NaCl concentrations (more than 80 mmol/L) induced sulforaphane production in different broccoli cultivars, while low NaCl concentrations were ineffective.

Nevertheless, NaCl treatment leads to a decline in yield, GS content, and middle-microelements in broccoli microgreens. Esfandiari et al. [10] reached similar conclusions. They showed that high NaCl solution concentrations decreased the germination and fresh weight of broccoli sprouts that were cultivated under light, as well as GS, especially glucoraphanin, glucoerucin, glucoiberverin, and gluconapin. However, Guo et al. [31] showed that higher NaCl concentrations increased the content of glucoraphanin. The discrepancies may have arisen from different cultivation methods: the broccoli seed in the Guo et al. [31] study was sown on a filter paper upon vermiculite, and grown in an incubator in darkness. The broccoli microgreens cultivar used in our study was rich in P, Ca, and K contents, which were inconsistent with previous studies [32]. The contents of Ca and Mg decreased under NaCl treatment, a finding that is similar to the results of Hassini et al. [33].

More than alleviating yield decline caused by NaCl stress, adding CaSO_4_ improved yield and increased the leaf area of broccoli microgreens (Figure 1). Leaf expansive growth and structural growth depends on hydraulics and carbon metabolism [34]. Turgor could be a candidate for the coupling among cell wall metabolism, cell wall mechanics, and the hydraulic control of plant cell growth [34]. Salinity can cause higher leaf turgor [35]; consequently, the pressure caused by turgor transmits stresses to the cell wall, which stretches irreversibly [36]. Moreover, it has been reported that Na^+^ causes softening of the cell wall, and Ca^2+^ signaling participates in maintaining cell wall integrity during NaCl stress. According to the “pectate cycle” mechanism, the cell wall must recruit a large amount of Ca, in order to tighten new wall matrix during the process of wall expansion [37]. Therefore, it can be inferred that the reason for growth promotion appearance under the combined treatment is that Na^+^ softened the cell wall, and the high NaCl concentration provided turgor for cell growth; meanwhile, Ca^2+^ participated in maintaining cell wall integrity via signaling, and leading to new cell wall construction as a raw material.

Along with yield improvement and growth promotion (Figure 1), the combined treatment of NaCl and CaSO_4_ altered the nutritional traits of broccoli microgreens. Firstly, the broccoli microgreens were Ca-fortified from the addition of CaSO_4_, which was consistent with results from hydroponic tomatoes [38]. Secondly, the decline in the content of GS caused by NaCl treatment was impaired from adding CaSO_4_, and more than half of individual GS contents were consistent with the control. Notably, the content of 4-hydroxyglucobrassicin under the combined treatment of NaCl and CaSO_4_ was the highest among all treatments. These results were in accordance with Sun et al. [39], who found that 4-hydroxyglucobrassicin content nearly doubled in CaCl_2_ solution. Preharvest CaSO_4_ treatment was reported to improve the content of total GS and individual GS in broccoli microgreens [24], which was consistent with our data. Thirdly, NaCl treatment increased the sulforaphane content, while adding CaSO_4_; namely, the combined treatment showed a similar sulforaphane level with the CaSO_4_ treatment. Owing to Ca^2+^ signaling that contributes to plant physiological response regulation [40], and the negative effect of Ca^2+^ on the sulforaphane formation during the glucoraphanin hydrolysis process in vitro [41], we speculated that the positive effect of NaCl on the sulforaphane production was indirectly and/or directly inhibited by Ca. In the research by Guo et al. [24], the sulforaphane content of broccoli microgreens improved from CaSO_4_ (10 mmol/L) treatment, mainly due to an increase in glucoraphanin; meanwhile, an increase in glucoraphanin was not found in our study. Such a discrepancy may be due to the relatively lower CaSO_4_ concentration that we applied, which did not significantly improve the production of glucoraphanin (Table 1), in addition to the negative effect of Ca^2+^ on sulforaphane formation during the glucoraphanin hydrolysis process [41]. Finally, since the presence of Cl^−^ and Ca can help enhance nitrate reductase activity to assimilate nitrate into organic nitrogen compounds [42], the combined treatment showed the lowest nitrate content. Accordingly, our findings indicate that the addition of CaSO_4_ in NaCl solution alleviated the unfavorable effects induced by NaCl stress.

## 5. Conclusions

The results showed that NaCl stress increased the content of sulforaphane, but reduced yields and the content of GS; moreover, adding CaSO_4_ increased yields (about two times), and obtained Ca-fortified, low-nitrate, and normal GS and sulforaphane content in broccoli microgreens products. This study provides new perspectives on the roles of Na and Ca in leaf growth. Instead of applying multi-element fertilizers [43], the incorporation of CaSO_4_ into NaCl solution provides a potential economical and convenient strategy to increase yields for microgreen grower communities, and to enhance broccoli microgreens quality for consumer’s special needs.

## Figures and Tables

**Figure 1 foods-11-03485-f001:**
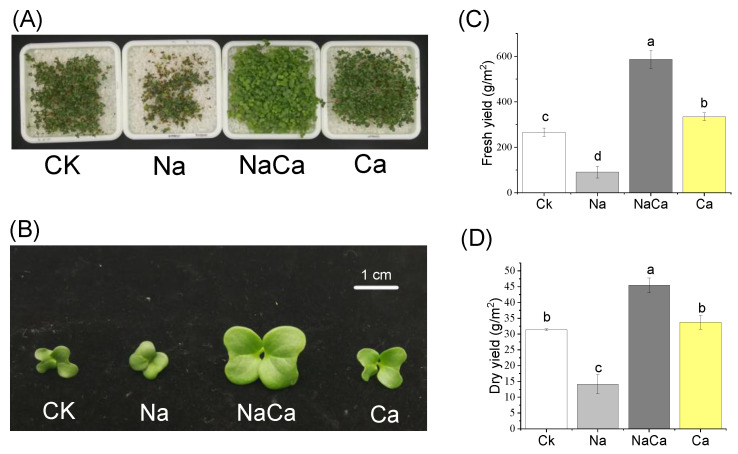
Fresh yield and dry yield of broccoli microgreens under different treatments. (**A**) Trays of microgreens under different treatments; (**B**) Leaves of microgreens under different treatments; (**C**) Fresh yield of microgreens under different treatments; (**D**) Dry yield of microgreens under different treatments. CK, distilled water; Na, 80 mmol/L NaCl; NaCa, 80 mmol/L NaCl + 1.39 mmol/L CaSO_4_; Ca, 1.39 mmol/L CaSO_4_; scale bar = 1 cm. Data are expressed as the mean ± standard deviation (SD), values labelled with different letters are significantly different (*p* < 0.05).

**Figure 2 foods-11-03485-f002:**
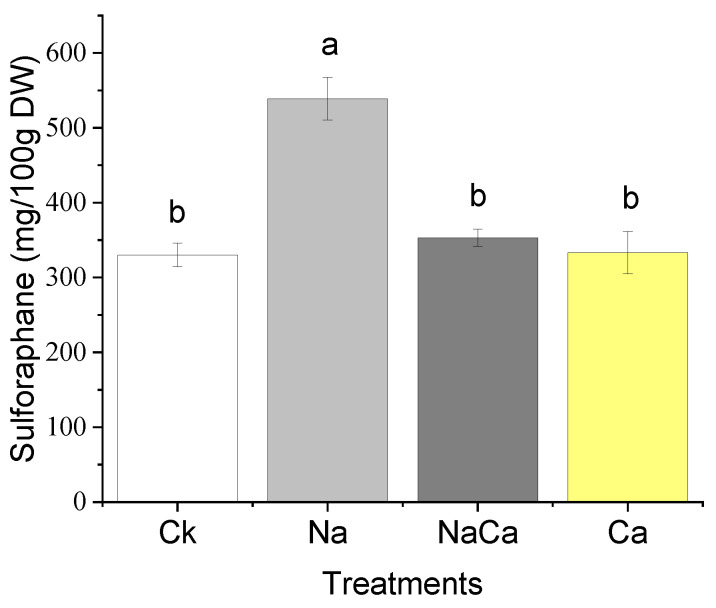
Sulforaphane content of broccoli microgreens under different treatments. CK, distilled water; Na, 80 mmol/L NaCl; NaCa, 80 mmol/L NaCl + 1.39 mmol/L CaSO_4_; Ca, 1.39 mmol/L CaSO_4_; DW, dry weight. Data are expressed as the mean ± standard deviation (SD), values labelled with different letters are significantly different (*p* < 0.05).

**Figure 3 foods-11-03485-f003:**
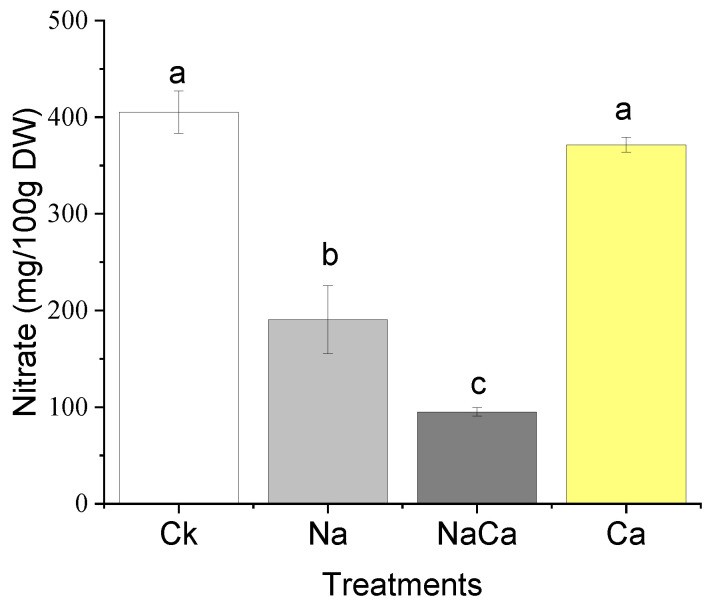
Nitrate content of broccoli microgreens under different treatments. CK, distilled water; Na, 80 mmol/L NaCl; NaCa, 80 mmol/L NaCl + 1.39 mmol/L CaSO_4_; Ca, 1.39 mmol/L CaSO_4_; DW, dry weight. Data are expressed as the mean ± standard deviation (SD), values labelled with different letters are significantly different (*p* < 0.05).

**Table 1 foods-11-03485-t001:** Glucosinolates composition and content (μmol/100 g dry weight) in shoots of broccoli microgreens under different treatments.

Treatments ^1^	CK	Na	NaCa	Ca
Glucoiberin	252.5 ± 14.6 ab	70.97 ± 4.79 c	195.7 ± 48.1 b	324.0 ± 67.6 a
Progoitrin	1252 ± 252 b	451.1 ± 16.2 c	988.2 ± 180 b	1887 ± 129 a
Signirin	246.2 ± 75.6 a	89.74 ± 4.57 b	200.8 ± 30.0 a	310.1 ± 19.9 a
Glucoraphanin	1944 ± 7.46 a	561.7 ± 25.5 c	1203 ± 203 b	2249 ± 167 a
Gluconapin	114.2 ± 8.44 b	42.72 ± 2.50 c	103.0 ± 23.1 b	150.4 ± 4.55 a
4-hydroxyglucobrassicin	402.8 ±22.5 b	227.7 ± 20.3 c	576.0 ± 74.0 a	522.0 ± 61.8 ab
Glucoerucin	1456 ± 34.2a	562.6 ± 9.51 c	1153 ± 118 b	1671 ± 124 a
Glucobrassicin	532.2 ± 7.38 b	166.7 ± 4.32 d	352.2 ± 84.2 c	664.9 ± 30.0 a
4-methoxyglucobrassicin	189.0 ± 5.49 a	63.30 ± 0.70 c	125.0 ± 17.8 b	141.0 ± 8.83 b
Neoglucobrassicin	88.09 ± 6.64 ab	52.58 ± 1.83 c	69.30 ± 17.1 bc	110.4 ± 8.95 a
Total glucosinolates	6478 ± 418 b	2289 ± 43.2 d	4966 ± 584 c	8031 ± 539 a

^1^ CK, distilled water; Na, 80 mmol/L NaCl; NaCa, 80 mmol/L NaCl + 1.39 mmol/L CaSO_4_; Ca, 1.39 mmol L-1 CaSO_4_. Data are expressed as the mean ± standard deviation (SD), values labelled with different letters are significantly different (*p* < 0.05).

**Table 2 foods-11-03485-t002:** Mineral content of broccoli microgreens under different treatments (mg/100 g dry weight).

Treatments ^1^	CK	Na	NaCa	Ca
Macroelements				
P	887.3 ± 15.9 a	876.8 ± 20.0 a	801.2 ± 18.9 b	878.5 ± 11.2 a
S	263.9 ± 2.32 c	262.5 ± 5.62 c	310.9 ± 7.18 b	368.1 ± 4.39 a
Na	127.8 ± 21.6 c	4729 ± 331 b	7514 ± 641 a	52.80 ± 20.2 c
K	488.1 ± 14.1 c	607.0 ± 10.7 a	505.1 ± 30.5 bc	581.1 ± 29.3 b
Ca	493.2 ± 20.3 c	360.9 ± 6.77 d	715.4 ± 16.1 b	1678 ± 41.8 a
Mg	302.7 ± 6.66 a	270.6 ± 8.23 b	228.9 ± 12.2 c	286.8 ± 3.43 ab
Microelements				
Zn	7.54 ± 0.46 a	6.61 ± 0.26 b	6.96 ± 0.34 ab	7.30 ± 0.07 ab
Fe	11.78 ± 1.32 a	13.23 ± 3.17 a	12.86 ± 2.58 a	12.72 ± 0.90 a

^1^ CK, distilled water; Na, 80 mmol/L NaCl; NaCa, 80 mmol/L NaCl + 1.39 mmol/L CaSO_4_; Ca, 1.39 mmol/L CaSO_4_; Data are expressed as the mean ± standard deviation (SD), values labelled with different letters are significantly different (*p* < 0.05).

## Data Availability

All data are contained within the article.

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
