# Peer review of "CaSO4 Increases Yield and Alters the Nutritional Contents in Broccoli (Brassica oleracea L. Var. italica) Microgreens under NaCl Stress"

_foods, 2022, doi:10.3390/foods11213485_

Round 1
Reviewer 1 Report
CaSO4 increases yield and alter the nutritional contents in broccoli (Brassica oleracea L. Var. italica) microgreens under NaCl stress
Manuscript is well written and contains very useful information. Introduction covers well the topic, all materials and methods are well described. Also results are presented clearly and discussion presents the effort to compare results with data published in literature. I can see only one problem in effort of results comparison. Authors mentioned several times in the text “dilution effects” (see e.g. lines 212, 215 and 231). If we look at Figures 2 and 3 results are presented as milligrams per 100 g. What represent these 100 grams? Fresh broccoli microgreens are true or dried ones? If we can compare such data with data of other authors or own data provided assisted by CaSO4 we need to compare data related on gram of dried material.
There is the reason why I recommend recalculating all data on gram of dried material, if possible. Then the comparison with other authors will be easier. "Dilution effects" can be neglected.
Author Response
Thank you for your careful evaluation of this manuscript.
Please see the attachment.

Reviewer 2 Report
I read with interesting your manuscript. I recommend this paper – after minor revision.
- add in Keywords: broccoli
- ln 60: mineral elements correct to : minerals (please provide them)
- M&M – reference to statistical analysis
- M&M - It appears that there was no quality assurance step (e.g. analysis of certified reference materials, recovery study) to validate the results of the concentrations data. Add informations (for minerals analysis)
- in table 2 ‘mineral elements’ change to: ‘minerals’, in table – divide macro- and microelements
- the conclusion should not be a summary. Make sure the conclusion is short and solid. Add a practical implications statement. An idea may be to synthetize in 3-5 bullet the key results of the study, evidences and recommendation. This improvement will increase clearness and readability. Add a practical implications statement
Author Response
Thank you for your kind comments and responded in a point-by-point manner.
Please see the attachment.
